# Saturated Transposon Analysis in Yeast as a one-step method to quantify the fitness effects of gene disruptions on a genome-wide scale

**Enzo Kingma, Floor Dolsma, Leila Iñigo de la Cruz, Liedewij Laan** * 

Department of Bionanoscience, Kavli Institute, Delft University of Technology, Delft, Zuid-Holland, The Netherlands

* L.Laan@tudelft.nl

**Data Availability Statement:** All SATAY sequencing datafiles from multiple biological and technical replicates of Saccharomyces cerevisiae

## Abstract

Transposon insertion site sequencing (TIS) is a powerful tool that has significantly advanced our knowledge of functional genomics. For example, TIS has been used to identify essential genes of *Saccharomyces cerevisiae*, screen for antibiotic resistance genes in *Klebsiella pneumoniae* and determine the set of genes required for virulence of *Mycobacterium tuberculosis*. While providing valuable insights, these applications of TIS focus on (conditional) gene essentiality and neglect possibly interesting but subtle differences in the importance of genes for fitness. Notably, it has been demonstrated that data obtained from TIS experiments can be used for fitness quantification and the construction of genetic interaction maps, but this potential is only sporadically exploited. Here, we present a method to quantify the fitness of gene disruption mutants using data obtained from a TIS screen developed for the yeast *Saccharomyces cerevisiae* called SATAY. We show that the mean read count per transposon insertion site provides a metric for fitness that is robust across biological and technical replicate experiments. Importantly, the ability to resolve differences between gene disruption mutants with low fitness depends crucially on the inclusion of insertion sites that are not observed in the sequencing data to estimate the mean. While our method provides reproducible results between replicate SATAY datasets, the obtained fitness distribution differs substantially from those obtained using other techniques. It is currently unclear whether these inconsistencies are due to biological or technical differences between the methods. We end with suggestions for modifications of the SATAY procedure that could improve the resolution of the fitness estimates. Our analysis indicates that increasing the sequencing depth does very little to reduce the uncertainty in the estimates, while replacing the PCR amplification with methods that avoid or reduce the number of amplification cycles will likely be most effective in reducing noise.

## Introduction

Measuring the phenotype of gene deletion mutants has been instrumental to our understanding of cell and evolutionary biology. In particular, the relation between genotype and fitness is

are available from the ArrayExpress database (accession number(s) E-MTAB-14476).

**Funding:** Funding from the European Research Council under the European Union's Horizon 2020 research and innovation programme (grant agreement 758132). The funders had no role in study design, data collection and analysis, decision to publish, or preparation of the manuscript.

**Competing interests:** The authors have declared that no competing interests exist.

a key element to move from descriptive to predictive evolutionary models. This mapping from genotype to fitness is typically conceptualized in the form of a fitness landscape. Importantly, the degree to which evolution can be predicted depends on the structure of this fitness landscape. Theoretical work has shown that the ruggedness of this landscape (that is, the number of fitness peaks) is an important feature that controls the predictability of evolutionary pathways [1–4]. However, the construction of empirical fitness landscapes remains challenging. Traditional methods to determine fitness are based on growth measurements of reconstructed mutants carrying the mutation of interest. These approaches are generally low-throughput, allowing only a handful of mutations to be assessed each time [5–9]. As the field of evolutionary biology is moving towards a more holistic view, there has been an increased demand in the past decades for techniques that allow evaluation of a large number of mutants in a single assay [10, 11].

While complete fitness maps at the resolution of single point mutations are still experimentally infeasible [12], considerable progress has been made in methods to analyze fitness in large libraries of gene disruption or deletion mutants. One of the earliest examples of these high-throughput methods is the Synthetic Genetic Array (SGA) which was developed for yeast [13–15] and bacteria [16]. A striking achievement of the SGA was the construction of a global map of genetic interactions between non-essential genes of the *Saccharomyces cerevisiae* genome [17–19]. However, the automated SGA workflow relies on robotics for library construction and fitness assays [15, 20, 21], which profoundly reduces its general accessibility due to associated costs. As a consequence, SGA has only been used sparingly and the majority of the reported fitness values are derived from a single mutant library. This is a crucial limitation of SGA, as it is known that fitness effects can strongly depend on genetic background. For example, a large scale study by [22] showed that an astonishing 17% of the annotated essential genes of budding yeast are dispensable in a different genetic context. Thus, methods to estimate the fitness effect of gene disruptions on a genome wide scale should ideally be easily applicable across different genetic backgrounds and environmental conditions.

Newer and more flexible techniques for high-throughput fitness measurements are based on pooled fitness assays, followed by next-generation sequencing to detect changes in mutant frequencies [23]. In Barcode sequencing (Bar-seq), mutant genomes are tagged with a unique nucleotide sequence that allows their identification after sequencing [24]. Read counts are then typically converted into a metric for fitness by calculating the log-frequency slope of each barcode measured between two timepoints [25–27]. Although the pooled assay of Bar-seq greatly facilitates the fitness assessment, it has the same issues as SGA with regard to the laborious steps required for mutant library construction [26]. In addition, the number of mutants that can be assessed in a single Bar-seq experiment is relatively low (typically between 1,000–5,000) due to the limited availability of unique barcodes [25–27]. An alternative to Bar-seq that allows simple *de-novo* generation of mutant libraries and which is not limited by barcode availability is transposon insertion sequencing (TIS). TIS methods utilize the ability of transposons to randomly translocate between different molecules of DNA to generate a library of gene disruption mutants [28–30]. Typically, transposon mutagenesis is efficient enough to produce libraries consisting of more than 100,000 mutants and a single library will often contain multiple mutants carrying disruptions at different locations in the same gene [29, 31, 32]. As mutants are identified based on the transposon insertion site by sequencing the transposon-genome junction, the genomic library preparation and bioinformatics analysis of TIS data is relatively complex. However, protocols have been developed that combine TIS with Bar-seq to simplify these steps [26].

Despite its experimental flexibility, TIS has only scarcely been used to generate quantitative fitness maps of gene disruptions on a global scale [29, 33, 34]. In essence, the approach of

estimating gene disruption fitness from read count frequencies is similar for TIS and Bar-seq. [29] have indeed used the log-fold change in read count frequency to quantify fitness from TIS data of *Streptococcus pneumoniae* with results that were in good agreement with measured growth rate of gene deletion strains. However, methods developed to estimate fitness for one version of TIS can in general not directly be applied to other TIS variants for the following two reasons. First, different variants of TIS may suffer from their own type of insertional bias that depends on the specific type of transposon system that is used [35, 36]. The ubiquity of insertion biases in TIS is becoming increasingly clear, as deep analyses are uncovering biases for transposons with a previously assumed uniform insertion profile [35, 37]. Second, some transposons used in TIS can only insert at specific nucleotide sequences [29, 38], while others can insert anywhere in the genome [31, 39, 40]. This difference in accessible insertion sites can lead to marked differences in the expected complexity (the number of different mutants present) of the TIS library. As a consequence, the typical number of reads acquired per mutant can vary substantially between different TIS protocols.

Recently, a TIS method named SAturated Transposon Analysis in Yeast (SATAY) has been developed for the yeast *Saccharomyces cerevisiae*. In their original paper [31], used SATAY to identify changes in the fitness contribution of genes in different genetic and environmental contexts through a one-on-one comparison of their transposon insertion density. While effective, this approach can only be used to identify fitness changes of the same gene across datasets, but not of different genes within the same dataset. In addition, it neglects possible subtle differences that are reflected in read count but not in the insertion density. Here, we describe an approach to generate quantitative fitness maps from read count data obtained from a single SATAY dataset (Fig 1) which is based on models frequently used to analyze count data from RNA-seq experiments [24]. We demonstrate that the inclusion of transposon insertions that are not visible through sequencing due to low abundance is crucial to resolve fitness differences in the lower end of the spectrum. Furthermore, we show that our method generates reproducible results across technical and biological replicate experiments, but has substantial differences with fitness and genetic interaction maps that have been produced with Bar-seq

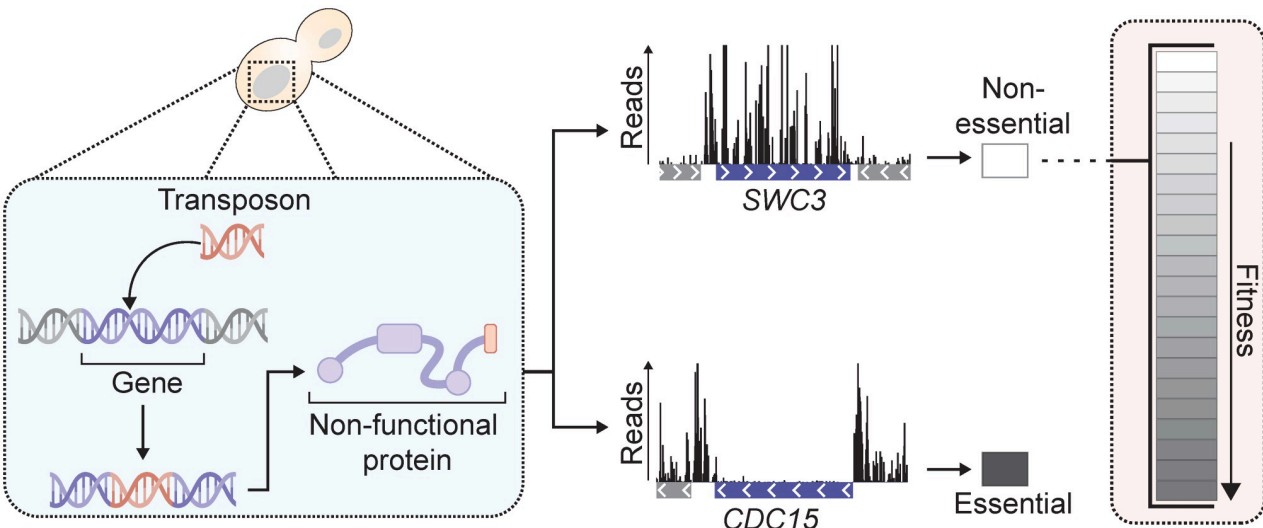

**Fig 1. Illustration of the aim of this work.** While existing methods for analyzing TIS data primarily categorize genes as either essential or non-essential based on transposon insertion density, our goal is to establish a method that quantifies the fitness effect of losing a non-essential gene on a more gradual scale.

and SGA by other studies. Finally, we give advice on possible adaptations of the experimental protocol that may improve the accuracy of the fitness estimates.

## Results

### Bias correction methods

**Insertions near gene edges are less likely to result in loss of gene function.** The read counts in TIS datasets are generally expected to correlate with mutant fitness, as fitter mutants will increase in frequency during the growth assay. However, in practice this correlation between fitness and read counts becomes obscured due to noise derived from stochastic growth trajectories and population sampling. An effective way to reduce the influence of these biological and technical noise sources on the fitness estimates is to average over several replicate measurements of the same mutant. Studies using TIS often assume transposon insertions to result complete loss of gene function [29]. Under this assumption, read counts obtained from different insertion sites within the same gene can be considered replicate measurements of the same gene deletion mutant and their averaging is justified. However, in other TIS systems it has been found that insertions near the 5' and 3' end of a gene are less likely complete loss of gene function [41]. Similarly, studies using SATAY have reported that genes considered to be essential can sometimes tolerate insertions close to the gene ends while central regions remain empty [31]. If this higher tolerance to insertions near gene ends is a general phenomenon affecting all genes, averaging the read counts of the entire coding region of a gene would create a bias in the fitness estimate.

We examined whether there exists a general trend of higher read counts for insertions close to gene edges that can be observed on a genome-wide level. To do so, we segmented the open reading frames of all annotated genes in the genome into 20 equally sized bins, such that each bin amounts to 5% of the coding sequence (Fig 2a). For each segment, we then calculated the average number of reads per transposon insertion that mapped to the respective segment. For genes annotated as non-essential, we observed that insertions within the first and last 10% of a gene tend to acquire more reads than insertions in the central 80% of the gene, although this effect is weak (Fig 2b). However, this non-uniformity was clearly visible for genes annotated as essential (Fig 2c). This difference between essential and non-essential genes is expected, as the fitness difference between a complete and partial knock-out should typically be larger for essential genes. Interestingly, the read count profile of essential genes shows that the bias towards higher read counts is strongest for insertions close to the stop codon. This likely reflects the mechanism for gene inactivation of the MiniDS transposon, which is based on creating gene truncations by introducing several early stop codons in the open reading frame [31]. Thus, insertions that lead to truncations close to the C-terminal part of the protein will often still allow the protein to (partially) retain its functionality.

Since insertions that do not lead to full gene knockouts invalidate the averaging over the read counts of different insertion sites, we decided to exclude all insertions that map within the first or last 10% of a coding region. However, even after removal of these insertion sites we find that the average read count distributions for essential and non-essential genes still overlap to some (Fig 2d). This indicates that using only the average read count as a metric for fitness does not allow us to distinguish essential from non-essential genes. A possible cause of this effect is that insertions in the essential regions of a gene remain unobserved because the corresponding mutants are lost before they can be sequenced. Neglecting these unobserved insertions means that our fitness estimates would be based completely on insertion sites that, for biological or technical reasons, have a higher read count than the typical insertion site. As a result, the fitness estimates become biased towards higher values and the ability to resolve fitness differences for

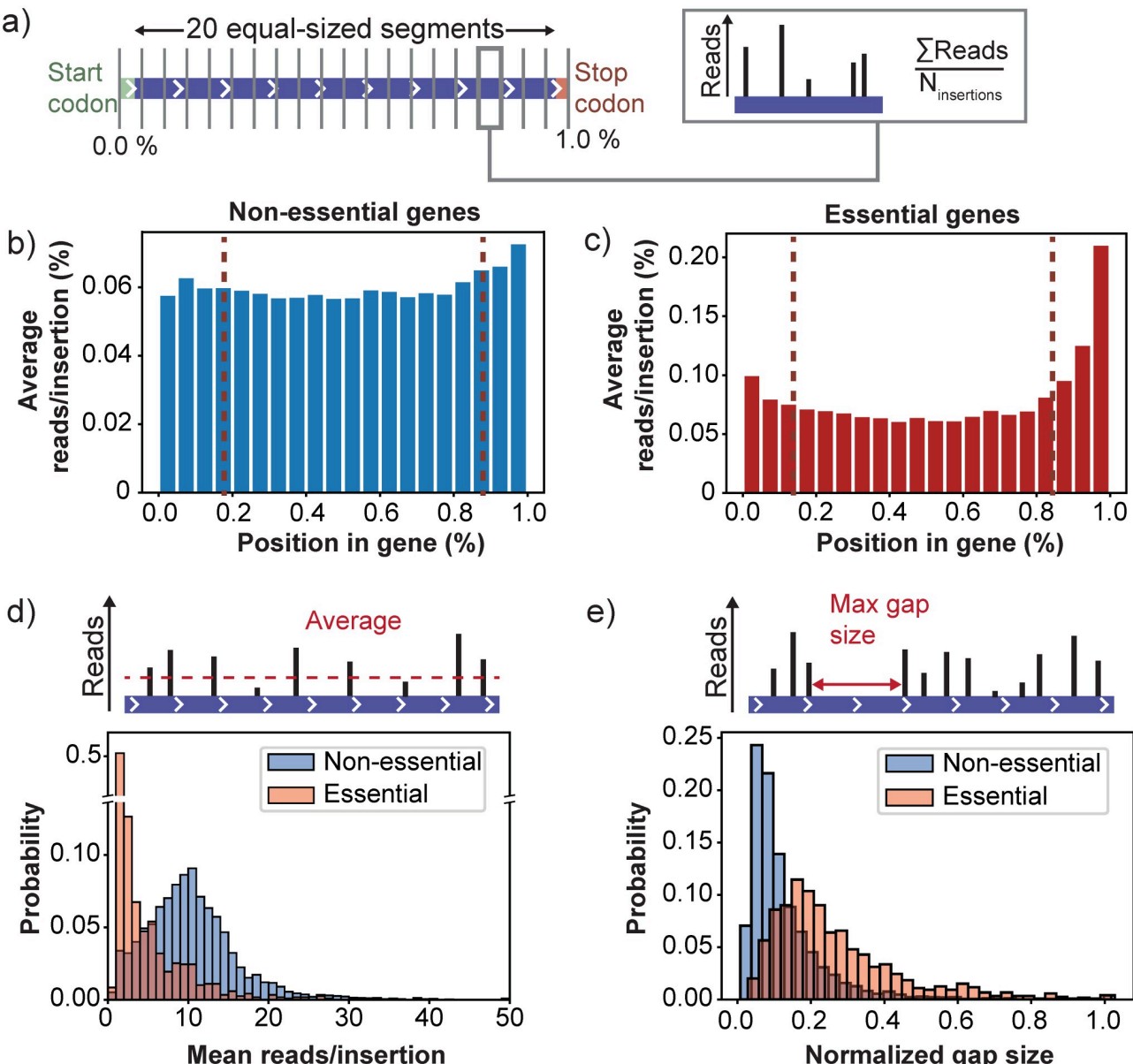

**Fig 2. Transposon insertions near the start and end of a gene are less likely to generate a gene knockout.** a) To identify a possible dependence of the read counts on the position of a transposon insertion within a gene, we split the open reading frame of each gene in 20 equally sized segments. Each segment therefore covers 5% of the coding region of a gene. For every segment we calculated the average read count per insertion site to obtain a profile along the coding region. b) Profile of the read counts against the position within the coding sequence, averaged over all non-essential genes. The profile shows that insertions near the start and stop codon of a gene obtain a slightly higher read count than insertions in the central 80% of the coding region. This effect is visibly stronger for genes that are annotated as essential (c), indicating that insertions near the gene edges do not cause a full gene knockout. d) The read count per insertion averaged over the central 80% of the coding region for essential and non-essential genes. While essential genes typically have a lower average read count per insertion site than non-essential genes, their distributions still overlap. e) Histogram of the largest span that appears free of insertions, expressed as a fraction of the gene's length. The distribution for essential genes has a tail towards longer insertion free spans (up to the full length of the coding region), while the spans are typically shorter for non-essential genes.

low-fitness mutants is lost. In support of the idea that unobserved insertion sites play a role in this bias, we find that the consecutive span of insertion sites that appear as unoccupied tends to be longer for essential genes (Fig 2e). Thus, essential genes are therefore ideally identified using other analysis methods, such as those based on the insertion-free span [42].

**Correcting for the preferential insertion of MiniDS at pericentromeric sites.** Mutants that are rare after the fitness assay, either because they grow poorly or have a high death rate, are likely to be lost during the sampling steps leading up to sequencing. Hence, the insertion sites associated with these mutants remain undiscovered in the final SATAY dataset (Fig 3a). While this loss of rare transposon mutants forms the basis for the identification of essential genes, it can distort the relation between mutant fitness and the read counts per insertion site. Specifically, including only insertion events that are observable from the sequencing data increases the susceptibility of the fitness estimate to gene disruption mutants that behave differently than the average. In addition, it puts the lower boundary for the average read count at 1, which severely reduces the number of distinct fitness levels that can be resolved for lower fitness values.

One approach to reduce these distortions is to include the undiscovered insertion sites in the fitness estimate as insertion sites with a read count of zero. This makes the fitness value of a gene dependent on the combination of the average read count per insertion site and the fraction of insertion sites that yield a read count larger than zero. To determine the number of insertion sites with zero reads for each gene, we use the global insertion density to infer an expected insertion rate. However, an issue with this approach is that the probability of insertion of the MiniDS transposon in SATAY depends on the distance of the insertion site to the centromeric regions of the chromosome. Because the MiniDS transposon preferentially inserts close to its excision site, the experimental design of SATAY has been reported to yield higher transposon densities in pericentromeric regions when compared to more distal regions. To verify the existence this centromere bias in our insertion data, we plotted the cumulative insertion count as a function of varying distances to the centromere (Fig 3b). This indeed revealed an enrichment of transposons in genomic regions that are closer than approximately 200 kb from a centromere. This effect was not the result of a lower density of essential genes in pericentromeric regions (S1 Fig). If left unaddressed, this bias would lead to an underestimation of the expected insertion rate in pericentromeric regions.

Ideally, a bias-corrected curve for the transposon insertion rate should follow the trend of the empirical data, while smoothing out any high frequency fluctuations. This is because such fluctuations are likely caused by the specific genetic content of the different genomic regions. We assessed two methods to model and correct for the centromeric bias: (1) a power-law and (2) a $4^{th}$ order polynomial (Fig 3c). While the plot of cumulative insertion count against distance seems to approach a power-law relationship, inspection of the residuals revealed that the fit systematically over- and underestimates different sections of the curve (Fig 3c and 3d). The increased flexibility of a $4^{th}$ order polynomial provided a better approximation for distances smaller than 200 kb, but began to fluctuate wildly for larger distances (Fig 3c and 3d).

Based on these findings, we decided to use the polynomial function to model the changes in the insertion rate up to distances 200 kb from the centromere while assuming a constant insertion rate beyond 200 kb. This assumption is based on the idea that the centromeric bias is a 'memoryless' feature such that the insertion rate is no longer depends on the distance to the centromere beyond a certain point. Furthermore, we note that the insertion rate at 200 kb inferred from the polynomial fit closely matched the insertion rate obtained when fitting a linear curve to the portion of the curve for distances of 200 kb and larger (Fig 3b and 3d). By taking the derivative of the polynomial fit with respect to the distance from the centromere, we obtain the global trend of changes in the insertion rate. Importantly, this estimate does not follow the local fluctuations in insertion rate that are likely caused by the specific properties of genes at different positions along the chromosome. Overall, the insertion rate varied between approximately 0.12 bp$^{-1}$ for regions close to the centromere to 0.067 bp$^{-1}$ for more distal regions. Because centromere bias should not play a significant role for regions very far from

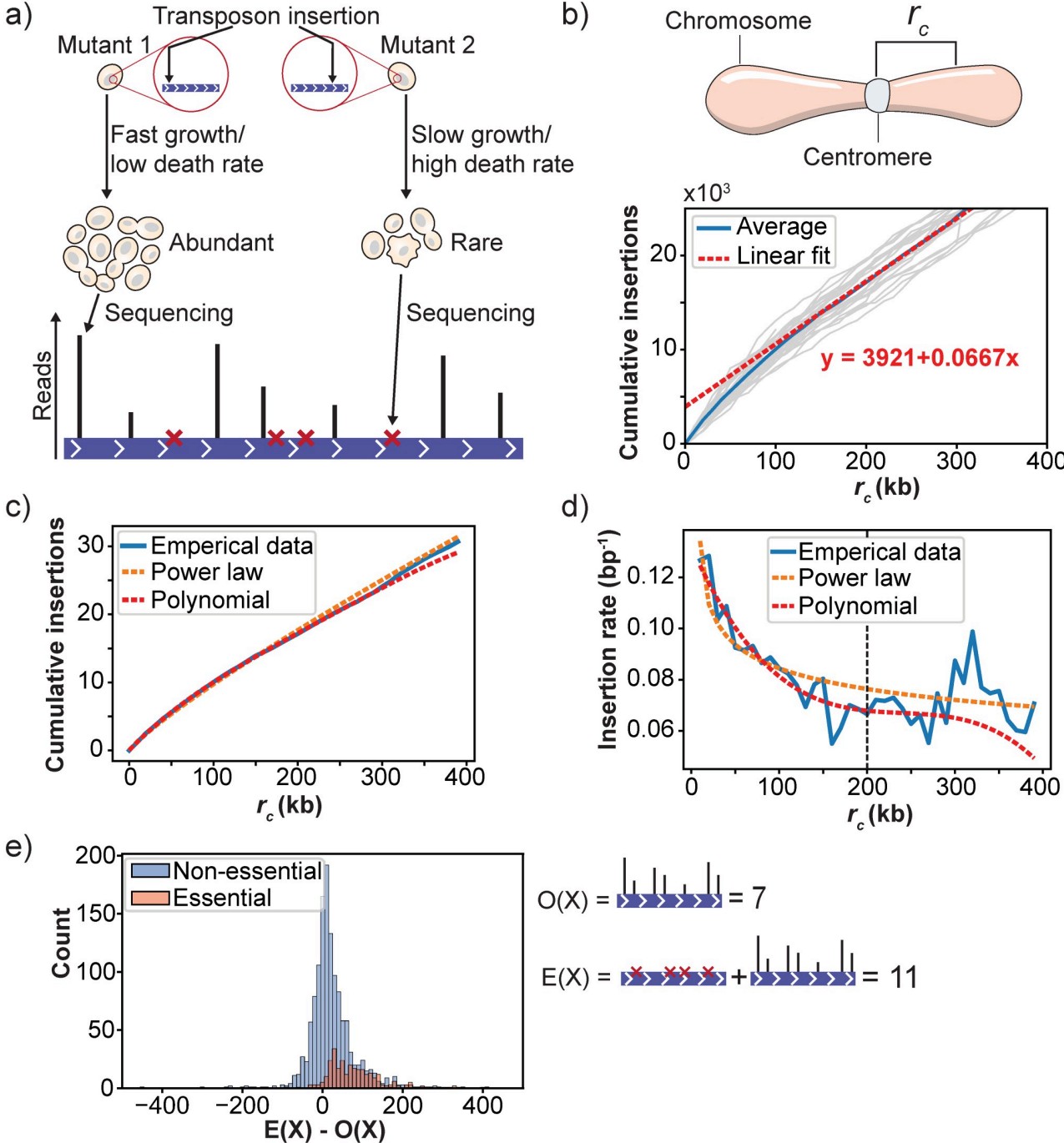

**Fig 3. Correcting for centromere bias to determine the expected transposon insertion rate of genes.** a) The goal of calculating the expected insertion rate is to estimate the number of insertion sites that produce no reads because the mutants were lost during the sampling of the population due to low abundance. b) The empirical insertion rate depends on the distance of a gene to the centromere. To visualize this bias we determined the number of transposons that mapped within a distance $r_c$ from the chromosome centromere for different values of $r_c$, as done previously by [31]. The plots of the cumulative insertions for the two halves of each individual chromosome are shown as grey lines, the averaged curve is shown as a blue line. The non-zero intercept of a linear fit of the portion of the average curve for distances $r_c > 300$ kb demonstrates the existence of the centromere bias in our dataset. c) The averaged cumulative plot for distances $R_c < 400$ kb was fitted with an exponential function and a 4th order polynomial. While the power law function does approximate the shape of the curve, it systematically under- and overfits different portions of the curve. Overall, the 4th order polynomial better approximates the curve. d) The approximated empirical insertion rate by the fitted power law and polynomial functions. The plot shows that while the approximation by the polynomial function is better for $r_c < 200$ kb, the polynomial starts to oscillate for larger distances. e) The difference between the expected (E(X)) and observed (O(X)) transposon insertion rates for essential and non-essential genes. While the average difference is close to zero for non-essential genes, it becomes positive for essential genes.

the centromere, we artificially flattened the fitted curve by setting a constant rate for positions located further than 200 kb away from the centromere. In summary, we use the following equation to determine a bias corrected insertion rate λ:

$$\lambda(r_c) = \begin{cases} a_0 + a_1 r_c + a_2 r_c^2 + a_3 r_c^3, & \text{for } r_c < 2 \cdot 10^5 \\ \lambda(r_c = 2 \cdot 10^5), & \text{for } r_c \geq 2 \cdot 10^5 \end{cases} \tag{1}$$

Where $r_c$ is the distance from the centromere in base pair and $a_{0-3}$ are the coefficients obtained from least squares polynomial fit. Because we assume that the insertion rate remains approximately constant over the span of a gene's coding sequence, the expected number of insertions for a gene $g$ is calculated by multiplying the insertion rate with gene size:

$$E(X_g) = \lambda(r_{c-g}) \cdot L_g, \tag{2}$$

with $r_{c-g}$ the distance from gene $g$ to the centromere measured from the gene's center and $L_g$ the length of the gene in basepairs. The number of zero read count sites is then estimated to be equal to the difference between $E(X_g)$ and the number of observed insertions $O(X_g)$ when $E(X_g) > O(X_g)$ (Fig 3a). When the number of observed insertions exceeds their expected value, we conclude that no unobserved insertions sites exist for that gene. We find that this procedure is able to correct for the skew in the expected number of insertion sites that produce zero reads for genes close to the centromere (S2 Fig).

To determine whether our model of the expected insertion rate was effective at identifying the larger fraction of undiscovered insertion sites in annotated essential genes, we plotted the difference in the expected and observed insertion counts for all genes (Fig 3e). Here, we consider this difference to reflect the number of transposon insertions that remain undiscovered after sequencing. The plot showed that for nearly all essential genes the expected number of insertions is higher than the number of insertions that are found from the read data. For non-essential genes, the difference between expected and observed insertion counts follows a Gaussian-like distribution around zero that partially overlaps with the distribution for essential genes. Thus, including undiscovered insertion sites provides additional information on essentialty, although not sufficient to distinguish essential from non-essential genes. However, our results show that inclusion of these zero-read count sites to estimate fitness is crucial to resolve differences in the lower end of the fitness distribution.

## Estimating fitness from read counts in SATAY

**Calculating mean mutant fitness.** Estimating the fitness effect of gene disruptions based on SATAY data relies on the relation between the observed read counts and mutant abundance. Specifically, fitter mutants proliferate more rapidly than less fit mutants, increase their relative abundance in the population and finally produce more read counts. The challenge of determining fitness from SATAY data can therefore be formulated as estimating the fitness parameter $\mu_g$, which we will refer to as the growth rate, using the observed read counts $y_{g,i}$ at the insertion location $i$ within gene $g$. While we will refer to and treat $\mu_g$ as a growth rate, we note that the interpretation of $\mu_g$ is significantly different from the conventional meaning of a growth rate in biology. Typically, growth rate is measured as the rate of population expansion during the exponential growth phase. Hence, by proposing a relation between growth rate and read counts we implicitly assume that the population remains in the exponential phase for the complete duration of the fitness assay. In reality, the length of the different growth phases (lag, exponential, stationary) can vary substantially between genetic backgrounds. All these variations will impact mutant abundance and are implicitly incorporated in the parameter $\mu_g$ rather

than explicitly modeled. However, we treat $\mu_g$ as the rate at which the mutant would have grown *if* it had remained in the exponential phase. Importantly, we assume that the abundance of a mutant with fitness $\mu_g$ increases over time according to the Malthusian growth model:

$$N_{g,i}(t) = N_{g,i}(t=0) \cdot 2^{\mu_g t}. \tag{3}$$

Here, $N_{g,i}(t)$ denotes the number of mutant cells that carry a transposon insertion at position $i$ in gene $g$ and $t$ represents time. We use the notation $\mu_g$ rather than $\mu_{g,i}$ to emphasize that each gene deletion mutant can be characterized by a single growth rate. Mutant abundance after a growth period $t$ is inferred from the number of reads that map to the location $y_{g,i}$. The partial distortion caused by the sampling and PCR amplification steps before sequencing yields an additional noise term $\epsilon$ in the relationship between mutant abundance and read count:

$$y_{g,i} = N_{g,i} \pm \epsilon = N_{g,i}(t=0) \cdot 2^{\mu_g t} \pm \epsilon. \tag{4}$$

Since read count values are limited to non-negative integers, we expect the distribution of $y_{g,i}$ of a gene $g$ to be Poisson-like. The Maximum Likelihood Estimator (MLE) of a Poisson distribution is equal to the sample mean, giving:

$$\hat{y}_g = \frac{1}{n_g} \sum_{i \in g} y_i. \tag{5}$$

Where $n_g$ is the total number of transposons that are mapped to gene $g$ and $\hat{y}_g$ is the MLE estimator of the read counts $y_g$ that would be obtained for the gene deletion mutant in the absence of noise. We emphasize that $\hat{y}_g$ will only be the MLE estimator of $y_g$ if it is ensured that all $y_{g,i}$ in Eq 5 are different realisations of sampling the same mutant with deleted gene $g$. Using this estimator for $y_g$, we can rewrite Eq 4 to get the following expression for $\mu_g$:

$$\mu_g = \frac{\log_2 (\hat{y}_g)}{t}. \tag{6}$$

Here we have omitted the term $\log(N_{g,i}(t=0))$ that would appear in the equation by assuming that the probability of multiple transposons inserting at the same genomic location is generally low, such that $N_{g,i}(t=0) = 1$ and $\log_2(N_{g,i}(t=0)) = 0$. To be able to compare the fitness values obtained from different experimental repetitions, possibly with different growth times, the growth rates must be scaled to that of the non-mutated ancestral strain. As other studies have shown that the majority of gene deletions have only a minor effect on fitness, we take the median of the distribution of growth rates to represent the fitness of of the ancestral strain:

$$w_g = \frac{\mu_g}{\mu_{ref}} = \frac{\log (\hat{y}_g)}{\mu_{ref}} \text{, with } \mu_{ref} = \text{median}[\log (\hat{y}_g)], \tag{7}$$

with $w_g$ the scaled fitness value of a mutant with gene $g$ deleted. The additional benefit of the expression shown in Eq 7 is that it does not depend on time. Thus, our method to calculate fitness from SATAY data depends only on the average read counts per insertion site.

Notice that for some genes the fitness value can be negative. These genes contain an excess of zero read count sites, resulting in the average read count value $\hat{y}_g$ becoming smaller than one. Because many insertion mutants are lost during the sampling steps of SATAY, a negative fitness value is not necessarily caused by a higher death rate of the gene deletion mutant. Instead, it should be interpreted to mean that the low effective growth rate of the mutant is so low that it is frequently lost from the population after the sampling steps. As such, the abundance of genes that are ascribed a negative fitness value will depend on the size of the

population bottleneck and the difference in growth rate between the fastest and slowest growing mutants.

**Variance of the fitness estimates.**   The identification of fitness differences that are statistically significant requires a method to estimate the uncertainty of the found fitness values. In the previous section, we proposed that the read counts of insertions mapped in the same gene ($y_{g,i}$) follows a Poisson-like distribution. However, other studies have shown that the variance in read counts from independent replicates are generally overdispersed compared to the Poisson distribution [43]. This overdispersion is the result of additional noise caused by biological sources. Thus, if the insertions mapping to different sites within the same gene represent different biological replicates, we would expect to observe this overdispersion in our read counts. We tested for overdispersion by plotting the average read count per insertion site against the variance for all genes (Fig 4c) This mean-variance relationship revealed that our data indeed displayed overdispersion compared to a Poisson model. To account for this overdispersion, we decided to use a Negative Binomial to model count noise, which is frequently used in the analysis of RNA-seq data [44–48]. We used the following parameterization of the Negative Binomial which relates its mean and variance through the overdispersion parameter $\alpha$:

$$\begin{cases} E(Y_g) = \mu_g. \\ Var(Y_g) = \sigma_g^2 = \mu_g + \alpha\mu_g^2. \end{cases} \tag{8}$$

In this equation, $\alpha$ regulates the degree of overdispersion relative to the Poisson model: as $\alpha$ shrinks to 0, our model becomes equivalent to a Poisson. Hence, the term $\alpha\mu_g^2$ can conveniently be interpreted as the variance due to biological noise that is added to the sampling noise.

To estimate $\alpha$ we assumed that (1) it can be estimated independently from the distribution mean and (2) that all genes in a single data set share the same value for $\alpha$. Our assumption that $\alpha$ and $\mu$ can be determined independently is justified by the fact that our fitness estimates are based on a single library and all read counts are therefore conditional on the same sequencing depth. Under these conditions, the MLE estimator for $\mu$ will always be the sample mean, regardless of the value of $\alpha$. We use the following regression equation to estimate $\alpha$ using Ordinary Least Squares (OLS) regression of the empirical mean-variance relationship:

$$\frac{(y_{g,i} - \hat{y}_g)^2 - \hat{y}_g}{\hat{y}_g} = \alpha\hat{y}_g. \tag{9}$$

Where the value of the estimator $\hat{y}_g$ is found using Eq 5. An example of the resulting fit for the mean-variance relationship obtained with this regression model is shown in Fig 4c. To account for the possibility that, for biological reasons, some genes have a higher variance than what is found when using the trended dispersion fit, we take the maximum of each genes individual sample variance and the variance obtained from Eq 8 to prevent underestimation of the true variance:

$$Var(Y_g) = \text{Max}\left(\sigma_{g,trend}^2, \sigma_{g,sample}^2\right). \tag{10}$$

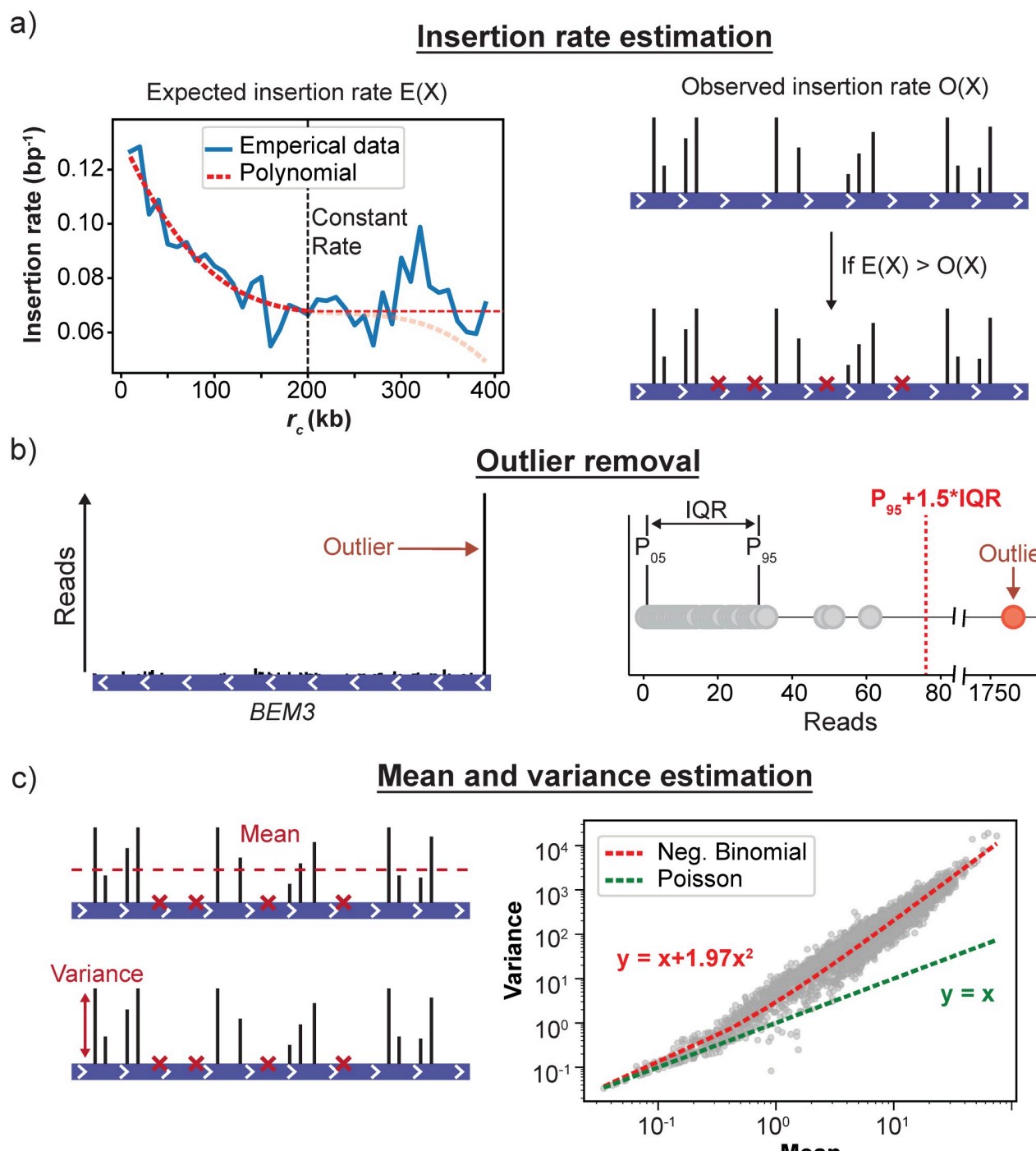

**Fig 4. The steps of the procedure to calculate fitness from transposon insertion data.** a) First, the expected number of insertion events is calculated for each gene, using the global insertion profile to correct for the centromere bias. The expected number of insertions is then compared to the observed number of insertions to determine the number of sites that have an insertion but produce no reads. These sites are included as sites that have zero read counts. b) After adding the zero read count sites, outliers are removed using the range between the 5th and 95th percentiles of the data. c) The mean and variance of the read counts at different insertion sites are used to determine the average and uncertainty, respectively, of the fitness of a gene deletion mutant. To provide a robust robust estimate for the variance, information is shared between genes by fitting the global mean-variance relationship with an overdispersed Poisson model. The resulting fit allows us to determine the variance based on the mean read count of a gene based on the assumption that this mean-variance relationship is a property of the dataset.

## Reproducibility of the fitness estimates

**Fitness values are reproducible across replicate SATAY experiments.** An important requirement for the applicability of SATAY for global fitness maps of gene disruption mutants is that the fitness estimates are reproducible between replicates of the same genetic background. To test this robustness, we created 11 replicate SATAY datasets using a wild-type strain of *Saccharomyces cerevisiae* from the W303 background. All 11 datasets are derived from the same wild-type transformed with a plasmid carrying the machinery to induce transposon mutagenesis [31, 49], but were subsequently split at different steps of the experimental procedure (Fig 5). Specifically, from the resulting transformation plate four colonies were picked (indicated as replicates B1-B4 in Fig 5) and used to generate four independent mutant libraries. For replicates B1 and B2, we sequenced the mutant library obtained after library expansion respectively six (replicates B1$_{T1-T6}$) and three (replicates B2$_{T1-T3}$ times (Fig 5). We refer to datasets obtained from individual colonies on the transformation plate as biological replicates, as the difference between these libraries are (although not exclusively) caused by differences in population composition and random birth/death events during library expansion. Alternatively, datasets obtained by re-sequencing the same library several times are referred to as technical replicates, as these exclusively contain the noise caused by random sampling during the sequencing process. Thus, this should allow us to distinguish the relative contribution of biological noise from the technical noise that results from limited sequencing of the samples.

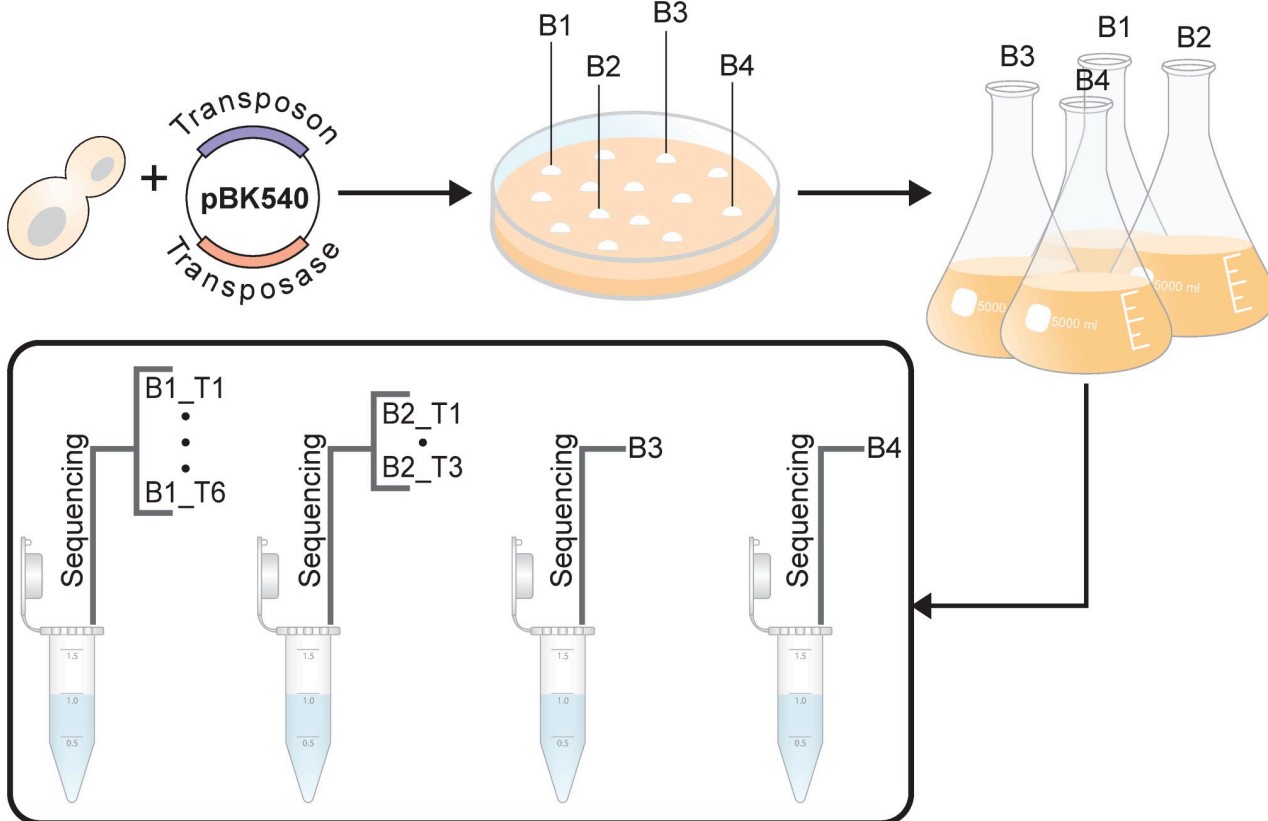

**Fig 5. Overview of the experimental procedure to determine the reproducibility of the fitness values across different replicate experiments.** Biological replicates B1-B4 are different clones of a single wild-type strain transformed with plasmid pBK549. For two of the biological replicates (B1 and B2), the extracted genomic DNA was sampled and sequenced multiple times, yielding the technical replicates B1$_{T1-6}$ and B2$_{T1-3}$.

To determine the reproducibility of our fitness estimates obtained from SATAY data sets using the procedure described in the previous sections, we compared the fitness values for each non-essential gene obtained from different technical (B1_1T vs. B1_T2) and biological replicates (B1 vs B2). We excluded genes for which the expected number of transposon insertions was less than five ($E(X) < 5$) on the basis that we have too little information from these genes to reliably estimate fitness. The results in Fig 6 show that fitness values are well

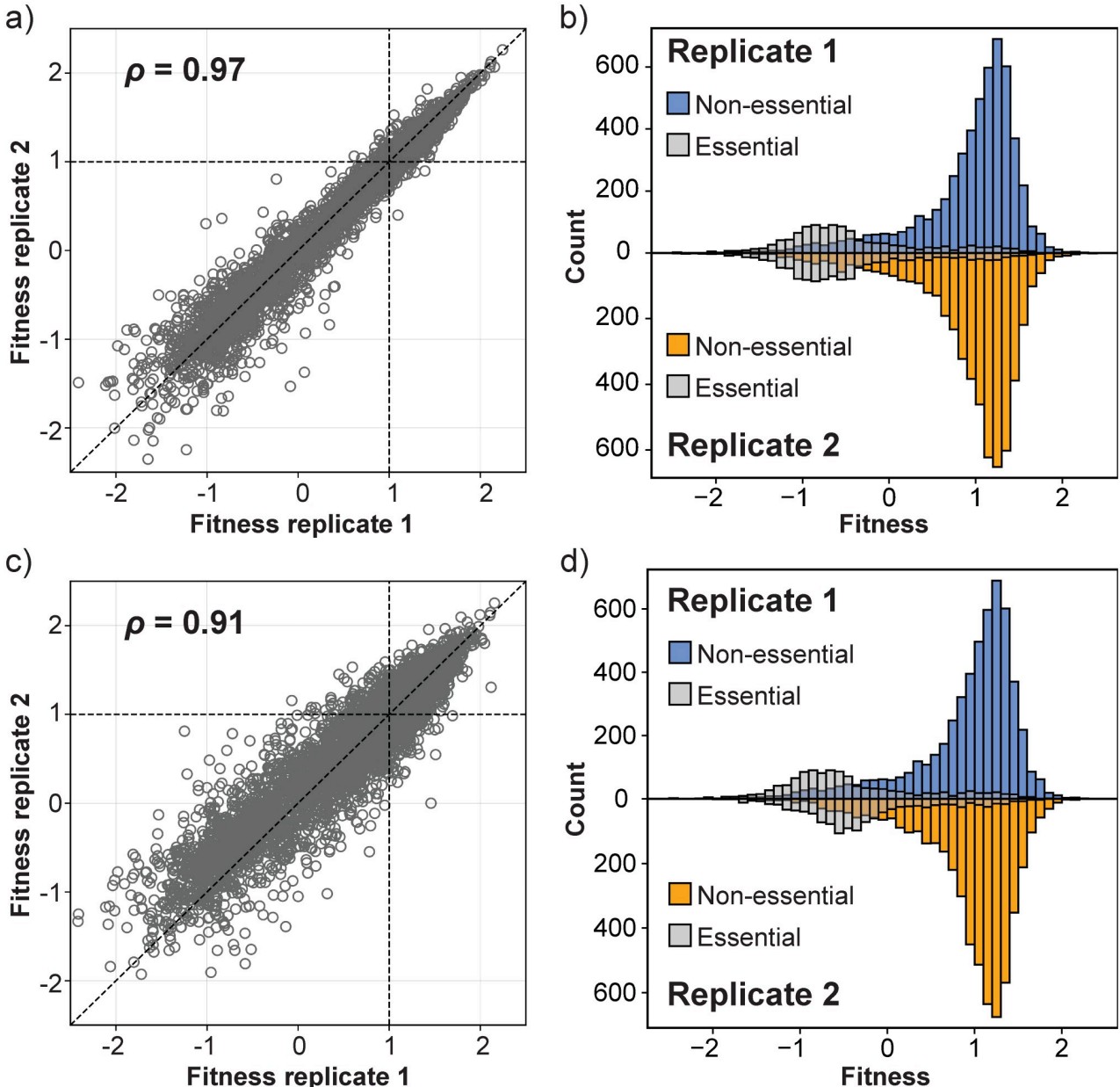

**Fig 6. Fitness estimates from SATAY are reproducible across replicate experiments.** a) The estimated fitness effect of gene disruptions for all genes of technical replicate 1 plotted against its estimated value in technical replicate 2. The identity lined (red dashed line) is shown as a reference for perfect correlation between the two replicates. b) The fitness distributions of technical replicate 1 (top) and technical replicate 2 (bottom). Essential genes are represented by gray transparent bars. c) Same plot as in (a), but for two biological replicates. d) The fitness distributions of biological replicate 1 (top) and biological replicate 2 (bottom). Annotations are the same as in (b).

correlated between technical ($\rho = 0.97$, Fig 6a) and biological replicates replicates ($\rho = 0.91$, Fig 6c), although the variance appears slightly higher between biological replicates.

In addition to a strong correlation of the fitness values between replicates, we furthermore find that for all samples the Distribution of Fitness Effects (DFE) contains a single peak with a skewed tail towards lower fitness values (Fig 6b and 6d). This is in agreement with the expected shape of the DFE when advantageous mutations are rare and most mutations have neutral or deleterious effects. We therefore conclude that fitness estimates based on read counts from SATAY data sets are reproducible across replicates of the same genetic background.

**Fitness estimates correlate poorly with reported estimates based on the yeast gene deletion collection.** Although we find that the fitness estimates strongly correlate between replicate experiments (Fig 6), we wanted to investigate whether we would be able to reproduce the fitness values and epistatic interactions reported by other studies. Particularly interesting is the study by [50], as they used Bar-seq to determine the fitness effects of gene disruptions. While fitness estimates are determined from read counts in both Bar-seq and SATAY, we need to account for the slight difference in the definition of fitness to compare our results to the results of [50]. Specifically, [50] use the following definition for the scaled fitness value $w_g$:

$$w_g = \left( \frac{P' P_{wt}}{P P'_{wt}} \right)^{1/t}. \tag{11}$$

Where $P'$ and $P'_{wt}$ are the mutant and wild-type frequencies, respectively, at the start of the competition experiment, $P$ and $P_{wt}$ are their frequencies at the end and $t$ is the number of generations over which the competition is evaluated. We therefore converted our fitness estimates to the same scale by setting $P$ equal to $\hat{y}_g$ (Eq 2), $P_{wt}$ equal to $\mathrm{median}[\hat{y}_g]$ and both primed variables equal to 1. The number of generations spent in library expansion is estimated to be approximately 10. Our adjusted fitness equation therefore becomes:

$$w_g = \left( \frac{\hat{y}_g}{median[\hat{y}_g]} \right)^{1/10}. \tag{12}$$

Fig 7a shows that there is a very poor correlation between the fitness values obtained by the two methods. In particular, the DFE obtained by [50] is much more centered around neutral fitness effects (that is, a fitness value of 1) and contains less spread compared to the DFE from SATAY. Thus, many genes that would be considered to have a near-neutral fitness cost when deleted based on the data by [50] are classified as deleterious in our fitness estimates from SATAY. These differences indicate that the fitness values obtained through the analysis of SATAY datasets may be condition specific and cannot be generalized without further considerations of experimental design.

Despite the low correlation with the data from [50], we wanted to examine if the fitness values obtained from SATAY could nevertheless be used to identify known epistatic interactions between genes. To this end, we compared the fitness values obtained from a wild-type strain with those from a *bem3Δ* mutant (Fig 7b). Interestingly, we do observe more variation between these two samples than we see for biological replicates of the same genetic backgrounds (Fig 6). In theory, genes that interact with *BEM3* will lie further away from the identity line than those that do not, since their fitness effect will either be suppressed (positive epistasis) or aggravated (negative epistasis) in the *bem3Δ* genetic background relative to their effect in the wild-type background. We annotated the positive and negative genetic interactors of *BEM3* that have been identified by an SGA screen [18] in Fig 7b to see if we could identify any correlation between the distance of the datapoint from the identity line and its annotation as a genetic

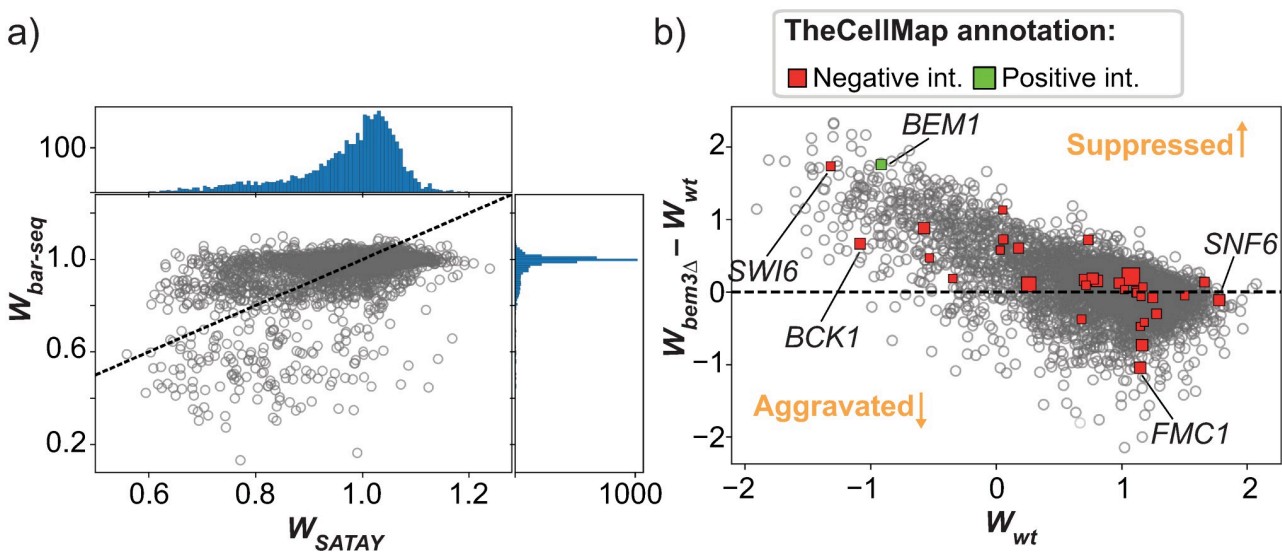

**Fig 7. The fitness values obtained from SATAY datasets only weakly correlate with the values reported by other studies.** a) Plot of the fitness values of gene deletion mutants reported by [50] against the fitness values obtained from a SATAY data set in this study. The top and side panels show the DFE densities. The identity line is shown for reference (dashed red line). b) Plot of the fitness values of gene deletion mutants generated using SATAY in a *bem3Δ* genetic background against the fitness values of the same gene deletion mutants generated in a wild-type (WT) genetic background (both from this study). Positive and negative genetic interactors of *BEM3*, as annotated by [18], are shown as green and red datapoints, respectively.

interactor. Note that although a different metric is used to calculate fitness (colony size), this SGA screen makes use of the same collection of gene deletion mutants [20] as [50] to screen for genetic interactions. We find that most annotated genetic interactors do not deviate more from the identity line than other, non-interacting, genes and are therefore not clearly identifiable. Hence, we would not be able to recover all of the genetic interactions of *BEM3* reported by [18] if we would base our analysis solely on the fitness values we obtain through our analysis of SATAY datasets. Therefore, similar to what we found for the fitness values obtained by different methods, the ability to predict epistatic interactions using the method for fitness estimation from SATAY data used here is inconsistent with other data sets.

## Decreasing the noise in the fitness estimates

While our fitness estimates are substantially different from those presented by [18, 50], our results in Fig 6 show that they typically vary little across replicate SATAY datasets. However, some applications may have specific requirements for the level of uncertainty in the fitness estimates. For example, if one wishes to resolve subtle differences in fitness it is desirable to design the experiment such that the highest possible accuracy is obtained. In these cases, it is useful to know how the experimental design can be modified to increase the accuracy and minimize noise sources. In this section, we survey different approaches to increase the resolution of fitness differences.

**Fitness resolution does not improve with increasing sequencing depth.** Intuitively, a simple way to improve the accuracy of the fitness estimates would be by increasing the sequencing depth. Specifically, deeper sequencing should allow better resolution of deleterious fitness effects [43], as low-abundance mutants are often lost when sampling the population. As sequencing depth increases, an important point is where all insertions are represented by at least one read. We refer to this as the saturation point of sequencing. When sequencing

saturation is reached, the information on mutant fitness is completely contained in the read counts and should no longer depend on the insertion density, as all unique insertion sites have been identified.

We attempted to determine the point of sequencing saturation for SATAY libraries by merging the data sets of technical replicates $B1_{T1-6}$ (see Fig 5). Because these technical replicates are resamplings of the same DNA sample, their merging results in a data set that is equivalent to deep sequencing of the sample in a single sequencing run. To estimate which portion of this merged data set needs to be sampled in order to obtain at least one read count for each unique transposon insertion event, we computationally sampled from this merged data set using a hypergeometric model. Fig 8a shows how the number of mapped transposon insertions changes as the number of reads in the subset increases. Although the rate of observing new transposon insertion does decrease as the total read count increases, we do not see clear signs

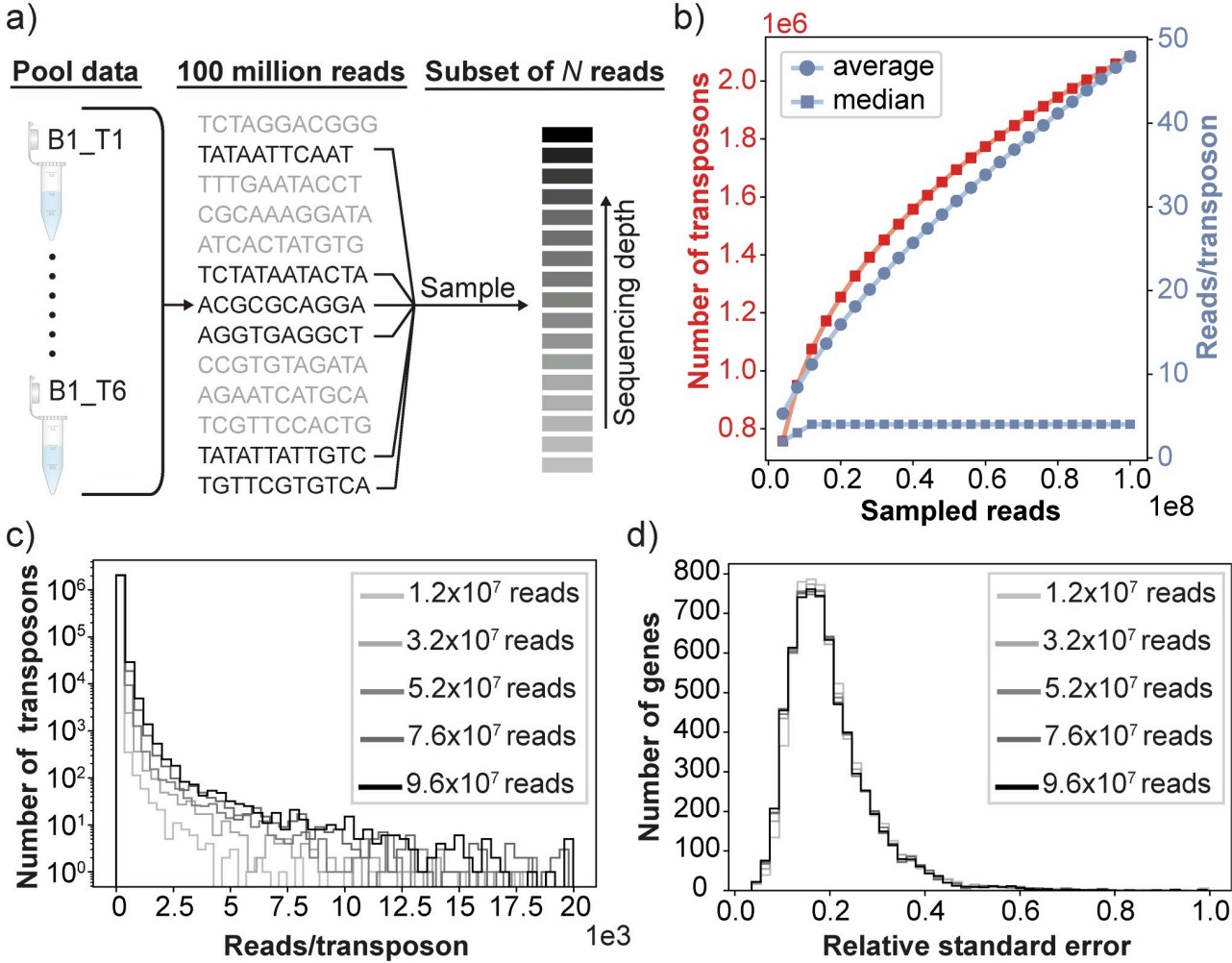

**Fig 8. Increasing sequencing depth does not improve the accuracy of fitness estimate.** a) The effect of sequencing depth on the accuracy of the fitness estimates was determined by pooling the experimental data from the six technical replicates B1_T1-B1_T6 and randomly sampling a subset of *n* reads without replacement from this pooled dataset. b) The number of observed independent transposon insertions (red) and the average and median read count per insertion site (blue) as a function of the number of sampled reads. c) The read count distribution for varying levels of the sequencing depth. The distribution has been cut off at a maximum value of $20 \times 10^3$ reads/transposon. d) The distribution of the relative standard error of the fitness estimates across different genes for different levels of sequencing depth.

of saturation despite the high sequencing depth of the full data set (approximately 100 million reads). In agreement with this we find that the median read count per transposon barely increases as the sequencing depth increases (Fig 8b), which indicates that the majority of unique insertion sites are still represented by only a single read. The inability to reach the saturation point limits the benefit of acquiring additional reads for the accuracy of fitness estimation. As is shown in Fig 8c, there is no clear improvement in the relative squared error of the mean of our fitness estimates when using data sets with a higher total read count. The saturation point may be reached when the sequencing depth is increased even further, but we consider going beyond 100 million reads to be impractical for any general application. Therefore, increasing the read depth of the sample appears to be an ineffective strategy to diminish the levels of sampling noise in the fitness estimates.

**Decreasing experimental noise.** We found that, while the correlation remains strong, the variance in the fitness estimates between biological replicates is larger than for technical replicates. Increased variability between biological replicates is expected, but depending its source, which may be either biological or technical, it might be possible to reduce it to increase the accuracy of the fitness estimates. with which fitness can be determined. To gain a better understanding of which process contributes to the increased noise levels of biological replicates, we compared the data sets obtained from replicates that were split at different steps of the SATAY procedure (Fig 9) Specifically, we looked at data sets from (1) separate colonies picked from the same transformation plate that were each processed through separate runs of library expansion and DNA extraction, (2) the same genomic DNA sample that was split before PCR amplification and sequencing and (3) the same PCR amplified DNA sample that was sequenced multiple times (Fig 5). This allows us to compare the contributions of biological noise, stochasticity introduced by PCR amplification and the sampling noise during sequencing to the variability we observe between biological replicates.

The degree of similarity between data sets was determined by comparing their mutant library composition based on the observed transposon insertion profile ($\Delta P$, Fig 9b) and the difference in read count obtained from transposon insertions mapped to the same genomic location ($\Delta R$, Fig 9b). Because the likelihood of acquiring a transposon insertion at exactly the same genomic location in independently generated mutant libraries is low and there is a limited accuracy of read mappping, we used a window of two basepairs within which transposon insertions were considered to match across replicate data sets (Fig 9b, left panel). The use of this window prevents the overestimation of differences in library composition, but should not significantly affect results based on the expectation that mutants carrying a transposon at nearly the same genomic location will have a similar defect. Based on the same rationale, an insertion site was only considered to be differently empty across two data sets if at least one of the insertions has a read count of two or more (Fig 9b, right panel).

The differences in read counts are plotted against the differences in insertion positions in Fig 9c for comparisons between data sets obtained from replicates split at the different steps of the SATAY procedure as depicted in Fig 9a. As expected, we find that the technical replicates that were split before sequencing (step 3 in Fig 9a) showed the lowest amount of variability in both $\Delta R$ and $\Delta P$, while the variability was higher for replicates that were split at earlier steps. Interestingly, while replicates split before library expansion (step 1) have the largest number of differential insertion sites ($\Delta P$), the difference in the average read count of matched insertion sites ($\Delta R$) was similar for replicates split before PCR amplification (step 2). This indicates that the differences in mutant library composition between replicate experiments has a relatively minor contribution to the differences in read count of matched insertions when compared to the effects of PCR amplification. The growth of a mutant during library expansion therefore appears to be insensitive to the presence or absence of other mutants. However, this

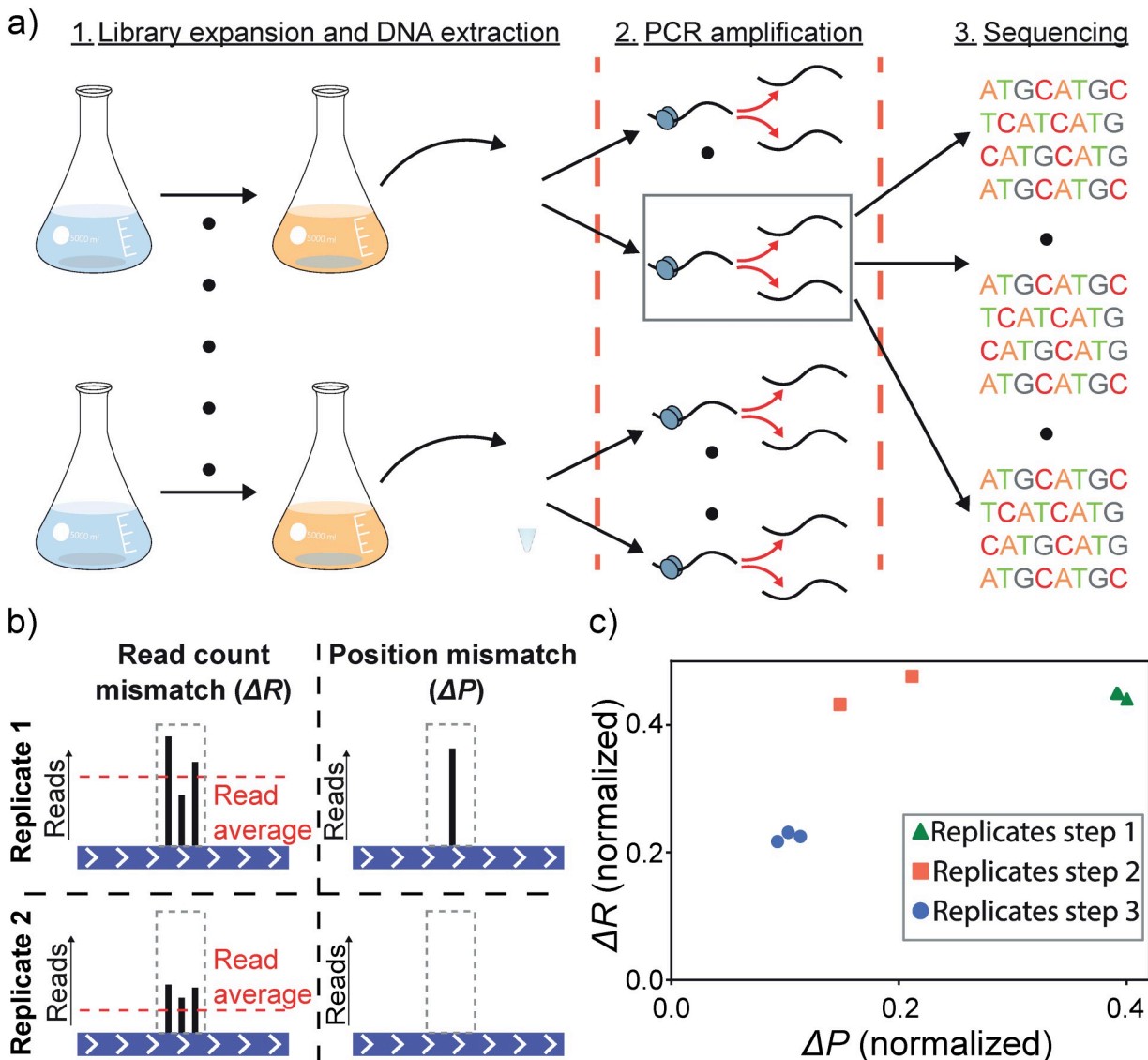

**Fig 9. The relative contributions of different noise sources during SATAY experiments.** a) Schematic representation of the different steps in the SATAY procedure at which different replicates were split. b) Procedure for comparing the difference in the insertion sites ($\Delta P$) and difference in read counts ($\Delta R$). If two replicates have non-zero read counts mapping to a position in the genome within 2 baspairs of each other (indicated by the grey box), the insertions are matched and their read difference is calculated (left panel). If one replicate has an insertion at a genomic location but no insertion is found within two basepairs of this location in the other replicate, this location is recorded as a difference in position (right panel). c) Plot of the differences in insertion sites and read counts between the different replicates. Replicates split before sequencing (step 3) show the smallest variation in both insertion positions and read counts. Replicates split before PCR amplification (step 2) have more variation in their read counts than samples split before sequencing. Splitting replicates before library expansion (step 1) creates the largest amount of variation in insertion sites between the replicates, but their read count difference at matched sites is similar to that of replicates split before PCR amplification.

insensitivity may depend on library complexity: When libraries are sufficiently complex, all possible mutant types that can be generated by transposon mutagenesis may be present in the population, although the precise transposon insertion locations that generate these mutants may differ between experiments. Thus, these observations indicate that PCR amplification adds a significant amount of noise to the observed read counts and adjustments to the

procedure that bypasses the requirement or reduces the amount of necessary PCR amplification during sample preparation may improve the accuracy of fitness estimation.

## Discussion

We developed a method to quantify the fitness of gene deletion mutants based on data from the transposon mutagenesis screen SATAY. Our approach differs from other methods that determine fitness from pooled assays in that our estimates are based on measurements of a single timepoint rather than the log-frequency change between two successive timepoints. Estimating fitness from a single timepoint required the assumption that all mutants are equally abundant at the start of library expansion (Eq 6). It is true that this assumption neglects the effect of altered growth conditions as the population is transferred from one media type to another during the SATAY protocol [51]. However, we anticipate that the impact of this media transfer on the growth trajectories is limited, as most transposition events occur during the saturated phase in induction media [49]. As a result, each mutant should only pass though a limited number of cell cycles before the transfer to the library expansion media. Moreover, it is important to recognize that sampling the population to measure mutant frequencies inevitably influences their growth trajectories. The distortion in growth caused by sampling can be particularly significant during early timepoints when mutant frequencies tend to be low.

To compare the fitness values obtained from different datasets, we rely on normalization to the median fitness of the population. This approach implicitly assumes that the majority of gene disruptions have a neutral effect on fitness. While we find that normalization to the median works well when the fitness distribution is unimodal, it is unlikely to produce reliable results for multimodal distributions. Conceptually, multimodal fitness distribution could occur when there are a small number of mutants that have a significantly higher fitness than the starting strain. Thus, the ability to compare fitness values from different datasets will depend on the fitness of the different genotypes present in the mutant library. This limitation has been addressed by [52], who showed that while providing accurate fitness ranks, fold-enrichment measures can yield biased proxies for fitness. Although computationally more intensive, approaches that explicitly model the growth trajectories of mutants during library expansion may be a robust alternative for normalization of multimodal distributions [53].

Gene deletion mutants from pre-constructed gene deletion libraries have been found to contain secondary mutations that compensate for the fitness defects caused by deleting a gene [20, 54, 55]. In SATAY, mutant libraries can be easily generated *de novo* for each experiment, making the fitness estimates less susceptible to the accumulation of secondary mutations. However, the complex relation between read count and fitness does increase the risk of introducing bias in the fitness values for the following reasons.

First, gene disruptions resulting from transposition may not always be equivalent to a full gene deletion, which can lead to underestimation of the fitness effect of a gene deletion. We indeed observed that insertions close to the 3' and 5' ends of a gene are often associated with a higher read count (Fig 2a and 2b). Excluding insertion events that map close to the gene edges is a relatively simple method to correct for this bias. However, it has been shown that the fitness effect of a transposon insertion also depends on the specific protein domain that it affects [31, 56]. The presence of multiple domains in an open reading frame therefore causes a more complex relation between read count and the position of an insertion within a gene. Correcting for such domain effects requires the grouping of insertions according to which domain they affect. While this would allow a more refined mapping of fitness based on domain-wise rather than a gene-wise fitness values, the spatial clustering of insertions based on their read count value is a complex problem that is beyond the scope of this study. Hence, the method

presented here does not account for these domain effects the reported fitness values should be interpreted as an average of the domain-wise fitness.

Second, using the average read count over all insertion sites within a gene as a metric to determine fitness makes the fitness estimate dependent on the mapping accuracy of the reads. Misalignment of reads derived from the same transposon insertion event causes them to become spread out over a larger genomic region. This leads to lower fitness values, as the reads mapping to a gene will now be divided by a larger number of transposons to obtain the average. Because the miniDS transposon used in the SATAY screen can insert at any basepair, it has a relatively higher susceptibility to this artefact compared to other TIS methods that use transposons that can only insert at specific nulceotide sequences. A possible sign that mapping accuracy plays a role in our data sets is the fact that we are unable to reach the point of sequencing saturation of the library, where each unique transposon insertion is represented by at least one read (Fig 8b and 8c). In addition, the number of identified unique transposon insertions is for a given total read count is systematically higher than what is reported by other studies using SATAY [31, 49]. One way to address the misalignment of reads is to discard reads with an alignment score below a certain threshold. However, this may disproportionately affect genes with a coding sequence that does not allow accurate alignment (such as those with repeated regions) which can artificially lower their read count.

We have shown that the method to estimate fitness presented in this chapter generates reproducible results across replicate SATAY experiments performed in the same genetic background. However, there was only a weak correlation with the fitness values reported by other studies and we were unable to retrieve epistatic interactions between genes that have been annotated based on SGA screens. We provide two explanations for this lack of consistency. First, we compared our results to the fitness values that were obtained from a pre-constructed gene deletion collection [18, 50, 57]. In pre-constructed gene deletion collections, mutants usually pass through several rounds of replication after their construction before they are subjected to a fitness assay. This gives the opportunity for secondary mutations that can potentially mask the defects caused by the primary gene deletion to set in the population. It has indeed been shown that several mutants in the yeast gene collection harbour such compensatory mutations [54, 55]. Libraries created with SATAY are less likely to suffer from secondary mutations because the fitness assay usually follows directly after library creation. As a consequence, fitness values based on measurements of the yeast gene deletion collection may overestimate the number of neutral genes compared to results obtained with SATAY, which is in agreement with our observation that the DFE obtained by [50] is much narrower than ours (Fig 7a). The second explanation is that the dissimilarities are caused by the different genetic background used by other studies (S288C [18, 20, 50], whereas we have used W303). Because the fitness effect of a gene deletion can depend on the allelic status of genes elsewhere in the genome, the observed inconsistencies may simply reflect a fundamental difference of the fitness landscape of the two genetic backgrounds. It is difficult to determine the extent to which the fitness landscape can be expected to diverge between genetic backgrounds, as only few studies have assessed fitness on the genome wide scale and most of these are based on the same collection of gene knock-outs. However, comparative studies of the genetic interaction maps of different species have shown that their structure can strongly vary [58], although some level of conservation does appear to exist [59–61].

In conclusion, the method we present here allows reproducible quantification of fitness from the read count data obtained from SATAY experiments. We hope that the considerable differences between our results and fitness and genetic interaction maps produced with other methods will spark initiatives to determine the amount of variation in these maps across genetic backgrounds.

## Methods

### Strains

All strains used in this study are of the W303 genetic background and are derived from a single parental strain (see S1 Table). The parental strain was made heterothrophic for adenine (*ade⁻*) by replacing the *ADE2* gene with the *URA3* marker using homologous recombination, followed by counterselection against the *URA3* marker with 5-fluoro-orotic acid to obtain a clean gene deletion. *bem3Δ* and *nrp1Δ* strains were created using homologous recombination to replace the endogenous genes with the natMX4 and hphMX4 cassettes, respectively. Strains were stored at -80˚C as frozen stocks in 40% (v/v) glycerol.

### Media

Standard culturing and growth assays were performed in YPD (10g/L Yeast extract, 20 g/L Peptone, 20 g/L dextrose), SC (6.9 g/L Yeast nitrogen base, 0.75 g/L Complete supplement mixture, 20 g/L dextrose). For *ade⁻* strains, standard growth media was supplemented with 20 mg/L adenine just before incubation. Liquid media for the preculture and induction steps of SATAY were prepared according to the recipe in S2 Table. After preparation, the media was filter sterilized using Rapid-Flow Sterile Disposable Filter Units (Nalgene) and stored at 4˚C until used. Liquid media for the reseed step of SATAY was prepared by autoclaving 2.6 L of MiliQ water in a 5 L flask. 400 ml of a 7.5X concentrated solution of the nutrients was prepared separately and filter sterilized. To prevent the degradation of media components, this concentrate was stored in the dark at 4˚C until used. On the day of reseed, the concentrate was aseptically added to the 5 L flask containing 2.6 L of MiliQ water and mixed. Solid media was prepared by adding 20 g/L agar and 30 mM Tris-HCl (pH 7.0) to the liquid media recipe and autoclaving the mixture for 20 minutes at 121˚C. 20 mg/L adenine was aseptically added after autoclaving, unless plates were intended to be selective for adenine auxotrophy.

### SAturated Transposon Analysis in Yeast (SATAY)

**Library generation.** SATAY libraries were generated based on the procedure described by [49], which is a modification of the original protocol [31] to allow transposition to occur in liquid media. *ade⁻* cells were transformed with plasmid pBK549 [49], which was a kind gift from Benoît Kornmann, according to a lithium acetate transformation protocol [62]. To screen for clones transformed with the intact version of plasmid pBK549 (see [51] for details on the different species of pBK549), 12–24 colonies were picked from the transformation plate, re-streaked on fresh SD-ADE and SD-URA plates and incubated for 3 days at 30˚C. For clones that showed full growth on SD-URA plates while producing a small number of colonies on SD-ADE plates, cells were scraped from the SD-URA plate and used to inoculate 25 ml of preculture media (S2 Table) at an OD600 of 0.20–0.28. Precultures were grown on an orbital platform shaker at 160 rpm, 30˚C until the OD600 was between 5–7 (∼20h). The saturated precultures were used to inoculate 200 ml of induction media at an OD600 of 0.10–0.27 and grown for 52 hours to allow transposition to occur. The efficiency of transposition was monitored by plating samples of the liquid induction cultures on SD-ADE at T = 0 and T = 52 hours and scoring the number of colonies on these plates after 3 days of incubation at 30˚C. After 52 hours of induction, the resulting transposon mutagenesis libraries were reseeded in 3 liters of reseed media at an OD600 of 0.21–0.26. Typically, this meant that around 7 million transposon mutants were reseeded per library. Reseeded libraries were grown for 92 hours at 140 rpm, 30˚C. At the end of reseed, cells were harvested by centrifugation of the reseed cultures at 5000 xg for 30 minutes. Cell pellets were stored at -20˚C.

**Genomic DNA extraction.** A 500 mg frozen pellet was resuspended in 500 μl cell break-ing buffer (2% Triton X-100, 1% SDS, 100 mM NaCl, 100 mM Tris-HCl pH8.0, 1 mM EDTA) and distributed into 280 μl aliquots. 300 μl of 0.4–0.6 mm glass beads (Sigma-Aldrich, G8772) and 200 μl of Phenol:Chloroform:isoamyl alcohol 25:24:1 (Sigma-Aldrich, P2069) were added to each aliquot and cells were lysed by vortexing the samples with a Vortex Genie 2 at maxi-mum speed at 4˚C for 10 minutes. 200 μl of TE buffer was added to each lysate, after which the samples were centrifuged at 16100x g, 4˚C for 5 minutes. After centrifugation, the upper layer (∼400 μl) was transferred to a clean eppendorf tube. 2.5 volumes of 100% absolute ethanol was added to each sample and mixed by inversion to precipitate the genomic DNA. After pre-cipitation, the DNA was pelleted by centrifugation at 16100x g, 20˚C for 5 minutes. The super-natant was removed and the DNA pellet was resuspended in 200 μl of 250 μg/ml RNAse A solution (qiagen, Cat. No. 19101). the resuspended DNA pellets were incubated at 55˚C for 15 minutes to allow digestion of the RNA. After digestion, 20 μl 3M, pH 5.2 sodium acetate (Merck) and 550 μl 100% absolute ethanol was added to each sample and mix by inversion. DNA was pelleted by centrifugation at 16100x g, 20˚C for 5 minutes. Pellets were washed with 70% absolute ethanol and dried at 37˚C for 10 minutes or until all ethanol had evaporated. The dried pellets were resuspended in a total volume of 100 μl MiliQ water and the concentration of the genomic DNA samples was quantified on a 0.6% agarose gel using the Eurogentec Smar-tladder 200bp-10kb as a reference. Prepared DNA samples were stored at -20˚C or 4˚C until used.

**Library sequencing.** To prepare genomic DNA samples for sequencing, 2x2 μg of DNA from each sample were transferred to non-stick microcentrifuge tubes and digested with 50 units of DpnII and NlaIII in a total volume of 50 μl for 17 hours at 37˚C. After digestion, the restriction enzymes were heat-inactivated by incubating the samples at 65˚C for 20 minutes. Digestion results were qualitatively assessed by visualization on a 1% agarose gel stained with Sybr-Safe. Successfully digested DNA samples were circularized in the same tube using 25 Weiss units of T4 DNA ligase (Thermo Scientific, Catalog #EL0011) at 22˚C for 6 hours in a total volume of 400 μl. After ligation, the circularized DNA was precipitated using 1ml 100% absolute ethanol, 20 μl 3M, pH 5.2 sodium acetate (Merck) and 5 μg linear acrylamide (invitro-gen, AM9520) as a carrier. DNA was precipitated for at least 2 days at -20˚C. Precipitated DNA was pelleted by centrifugation for 20 minutes at 16100x g at 4˚C and washed with 1 ml of 70% ethanol. After washing, the DNA was re-pelleted by centrifugation for 20 minutes at 16100x g at 20˚C, the supernatant was removed and pellets were dried for 10 minutes a 37˚C. Each dried pellet was resuspended in water and used as a template for 20 PCR reactions of 50 μl.

For samples sequenced on the Illumina HiSeq platform (yLIC136), PCR amplification of the transposon genome-junctions, sequencing and sequence alignment were performed by Agnès Michel and Benoît Kornmann (Oxford). For samples sequenced on the Illumina Nova-Seq 6000 platform (yWT01a, yLIC137), the transposon-genome junctions were amplified using the barcoded primers 1 and 2 (S3 Table) for DpnII digested DNA or primers 3 and 4 (S3 Table) for NlaIII digested DNA on a thermal cycler (Bio-Rad C1000 Touch) with the block settings shown in S4 Table. PCR amplified samples were purified using the NucleoSpin Gel and PCR cleanup kit (Macherey-Nagel) and quantified on the NanoDrop 2000 spectropho-tometer (Thermo Scientific). For each sample, equal ratios (w/w) of DpnIII and NlaII digested DNA were pooled. Library preparation and sample sequencing were performed by Novogene (UK) Company Limited. Sequencing libraries were prepared with the NEBNext Ultra II DNA Library Prep Kit, omitting the size selection and PCR enrichment steps. Libraries were sequenced using Paired-End (PE) sequencing with a read length of 150 bp.

**Sequence alignment.** FASTQ files obtained from the HiSeq platform were analyzed by Agnès Michel and Benoît Kornmann (Oxford) using their in-house pipeline for sequence processing and alignment [51]. FASTQ files obtained from the NovaSeq 6000 platform were demultiplexed into DpnII and NlaIII digested DNA samples based on the barcodes introduced during PCR amplification. Read pairs with non-matching barcodes were discarded. After demultiplexing, the forward read of each read pair was selected and the sequences upstream of primer 688_minidsSEQ1210 [31] and downstream of the DpnII (GATC) or NlaIII (CATG) restriction site were trimmed. All demultiplexing and trimming steps were executed with BBduk integrated into a home-written pipeline written in Bash. After trimming, the forward reads were aligned to the S288C reference genome (version R64–2-1_20150113) with the Transposonmapper pipeline ([63], version v1.1.4) using the following settings:

- Data type: 'Single-end'

- Trimming software: 'donottrim'

- Alignment settings: '-t 1 -v 2'

## Analysis of essential genes

Genes were marked as essential based on their annotation on the Saccharomyces Genome Database ([64], accessed on 03/17/2006).

## Centromere bias correction

Centromere bias was estimated based on the global transposon insertion density profile (see Fig 3). The insertion density profile was fitted with a third-degree polynomial using the polyfit function from the Numpy package (version 1.21.5). The derivative of the fitted polynomial was determined with the polyder function from the Numpy package (version 1.21.5). To calculate the distance (in bp) of a gene to the corresponding chromosome centromere ($r_{c-g}$ in Eq 1), the coordinates of each centromere were obtained from the Saccharomyces Genome Database (accessed on 02/06/2022). Either the start or stop position of the centromere was used to calculate distance, depending on which was closest to the gene of interest. Because the transposon insertion count can only take on discrete values, the value of the expected number of transposon insertions in a gene ($E(X_g)$ in Eq 2) was rounded down to the nearest integer.

## Fitness and variance calculation

Fitness values were based on the mean read count of transposon insertions that mapped to the central 80% of the coding region of a gene. Reads that exceeded 1.5 times the 5–95 percentile range of the read distribution of a gene were classified as outliers and removed. The 5–95 percentile range was determined using the stats module available from the Scipy package ([65], version 1.7.3) If the number of insertion sites used to calculate fitness was less than 5 after outlier removal or if all remaining insertion sites had a read count of 0, the fitness value was set to undetermined and not used for comparisons with other datasets.

The variance of the fitness estimates was determined from the observed mean-variance relationship of all genes (Fig 4c). The mean-variance relationship was fitted with Eq 9 using the OLS function available from the Statsmodels module. The obtained variance estimates were used to calculate the standard error of the fitness values with the following equation:

$$SE_g = \frac{\sqrt{V_g}}{\sqrt{n}} \tag{13}$$

Where $SE_g$ is the standard error of gene $g$, $V_g$ is the variance of gene $g$ estimated according to Eq 10 and $n$ is the number of transposon insertions that have been mapped to the coding sequence of gene $g$.

## Genetic interactions of *BEM3*

Genetic interactions of *BEM3* were downloaded from TheCellMap.org ([19], accessed on 13/07/2022). A stringent cut-off was used for the significance level of negative (GI-score $< -0.12$, p-value $< 0.05$) and positive (GI-score $>0.16$, p-value $< 0.05$) genetic interactions. Interactions annotated as dubious or those that were reported to have been affected by a suppressor mutation were excluded. In addition, only those interactions that were derived from fitness measurements of gene deletion strains (and not conditional knockouts) were used.

## Supporting information

**S1 File.**
(PDF)

**S1 Fig. Essential and non-essential genes are distributed in a similar manner across the chromosome.** Plot of the number of essential and non-essential genes for a specified distance $r_c$ from the centromere. The distributions do not show a clear enrichment of essential genes over non-essential genes in pericentromeric regions.
(TIF)

**S2 Fig. A polynomial fit of the observed transposon density across the genome corrects for the pericentromeric insertion bias of the MiniDS transposon.** The difference between the expected ($E(X)$) and observed ($O(X)$) insertion density for all genes is plotted against the distance of the gene to the centromere. a) Using the global average to estimate the expected insertion rate results in a systematic overestimation of the insertion density of genes close to the centromere. This overestimation is visible as a skew towards negative values for genes close to the centromere. b) Using a polynomial fit of the observed insertion rate corrects for this skew.
(TIF)

**S3 Fig. Inclusion of unobserved insertion sites improves the distinction between low-fitness genes.** a) The fitness distribution of gene disruptions when unobserved insertions are excluded during the fitness estimation. In this case, the fitness estimates depend only on the average read count per insertion site. b) The fitness distribution of gene disruptions when unobserved insertions are excluded during the fitness estimation. The plot shows that the lower tail of the distribution has a larger spread when compared to (a), providing a better distinction between low-fitness genes. In this case, the fitness estimates of genes that have low fitness when disrupted depends on a combination of the average read count and the insertion density.
(TIF)

**S1 Table. List of used strains.**
(PDF)

**S2 Table. Liquid media used for SATAY library generation.** YNB: Yeast Nitrogen Base. CSM: Complete Supplement Mixture. Ura: Uracil. Ade: Adenine.
(PDF)

**S3 Table. List of used primers.**
(PDF)

**S4 Table. PCR protocol used for the amplification of the transposon-genome junctions.**
(PDF)

**S5 Table. Characteristics of the SATAY libraries used in this chapter.** Reads that map to the coding sequence of *ADE2* were excluded when determining the reported values.
(PDF)

## Acknowledgments

We thank Sergey Kryazhimskiy and Benoît Kornmann for their useful comments on this manuscript. We are grateful to Benoît Kornmann and Agnès Michel for their help with setting up the SATAY experiments. We thank Benoît Kornmann, Agnès Michel, Wessel Teunisse and Werner Karel Daalman for their discussion on the data interpretation and analysis.

## Author Contributions

**Conceptualization:** Enzo Kingma, Liedewij Laan.

**Data curation:** Enzo Kingma, Floor Dolsma.

**Formal analysis:** Enzo Kingma, Floor Dolsma.

**Funding acquisition:** Liedewij Laan.

**Investigation:** Enzo Kingma, Floor Dolsma, Liedewij Laan.

**Methodology:** Enzo Kingma, Leila Iñigo de la Cruz, Liedewij Laan.

**Software:** Enzo Kingma, Leila Iñigo de la Cruz.

**Supervision:** Enzo Kingma, Liedewij Laan.

**Validation:** Enzo Kingma.

**Visualization:** Enzo Kingma.

**Writing – original draft:** Enzo Kingma.

**Writing – review & editing:** Enzo Kingma, Liedewij Laan.

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
