## [Decision Letter · Decision Letter 0]

15 Jul 2024

PONE-D-24-01082Saturated Transposon Analysis in Yeast as a One-step Method to Quantify the Fitness Effects of Gene Disruptions on a Genome-Wide ScalePLOS ONE

Dear Dr. Laan,

Thank you for submitting your manuscript to PLOS ONE. After careful consideration, we feel that it has merit but does not fully meet PLOS ONE’s publication criteria as it currently stands. Therefore, we invite you to submit a revised version of the manuscript that addresses the points raised during the review process.

Please submit your revised manuscript by Aug 29 2024 11:59PM. If you will need more time than this to complete your revisions, please reply to this message or contact the journal office at plosone@plos.org. Please include the following items when submitting your revised manuscript:A rebuttal letter that responds to each point raised by the academic editor and reviewer(s). You should upload this letter as a separate file labeled 'Response to Reviewers'.A marked-up copy of your manuscript that highlights changes made to the original version. You should upload this as a separate file labeled 'Revised Manuscript with Track Changes'.An unmarked version of your revised paper without tracked changes. You should upload this as a separate file labeled 'Manuscript'.If applicable, we recommend that you deposit your laboratory protocols in protocols.io to enhance the reproducibility of your results. Protocols.io assigns your protocol its own identifier (DOI) so that it can be cited independently in the future. For instructions see: https://journals.plos.org/plosone/s/submission-guidelines#loc-laboratory-protocols. Additionally, PLOS ONE offers an option for publishing peer-reviewed Lab Protocol articles, which describe protocols hosted on protocols.io. Read more information on sharing protocols at https://plos.org/protocols?utm_medium=editorial-email&utm_source=authorletters&utm_campaign=protocols.

We look forward to receiving your revised manuscript.

Kind regards,

Bashir Sajo Mienda, PhD

Academic Editor

PLOS ONE

“funding from the European Research Council under the European Union’s Horizon 2020 research and innovation programme (grant agreement 758132).”

Reviewers' comments:

Reviewer's Responses to Questions

**Comments to the Author**

1. Is the manuscript technically sound, and do the data support the conclusions?

Reviewer #1: Yes

2. Has the statistical analysis been performed appropriately and rigorously? 

Reviewer #1: I Don't Know

3. Have the authors made all data underlying the findings in their manuscript fully available?

Reviewer #1: Yes

4. Is the manuscript presented in an intelligible fashion and written in standard English?

Reviewer #1: Yes

5. Review Comments to the Author

Reviewer #1: #Review Report#

Dear Editor.

Thank you for the opportunity given to me to review this manuscript titled “Saturated Transposon Analysis in Yeast as a One-step Method to Quantify the Fitness Effects of Gene Disruptions on a Genome-Wide Scale” for publication in your PLOS ONE Journal. I have reviewed the manuscript and found it quite interesting. The manuscript is well-composed. The dataset therein is technically sound. There is enough justification to affirm the originality of the work done; the numerical figures of the various analytical factors have been reported with a reasonable degree of accuracy. The aim of the work, as outlined in the introductory part and illustrated in Figure 1 of the manuscript has been sufficiently achieved. On the general note, the work has underscored the critical role of saturated transposon analysis in quantification of fitness effect of gene disruption on a genome-wide scale, which could undeniably help in advancing the knowledge of functional genomics, particularly in the construction of genetic interaction maps. In my opinion, the manuscript has met both the scientific and ethical standards for inclusion in the publication of scientific record. However, I have outlined few comments below that could help improve the quality of the manuscript.

Comments:

1. P2/36. L75. The word “the” Consider removing one.

2. P6/36. L235. The word figure should be written with initial capital letter to make it consistent with the other usage in the manuscript.

6. PLOS authors have the option to publish the peer review history of their article (what does this mean?). If published, this will include your full peer review and any attached files.

Reviewer #1: No

---

## [Author Response · Author response to Decision Letter 0]

4 Oct 2024

Comments from reviewer #1: Dear Editor. Thank you for the opportunity given to me to review this manuscript titled “Saturated Transposon Analysis in Yeast as a One-step Method to Quantify the Fitness Effects of Gene Disruptions on a Genome-Wide Scale” for publication in your PLOS ONE Journal. I have reviewed the manuscript and found it quite interesting. The manuscript is well-composed. The dataset therein is technically sound. There is enough justification to affirm the originality of the work done; the numerical figures of the various analytical factors have been reported with a reasonable degree of accuracy. The aim of the work, as outlined in the introductory part and illustrated in Figure 1 of the manuscript has been sufficiently achieved. On the general note, the work has underscored the critical role of saturated transposon analysis in quantification of fitness effect of gene disruption on a genome-wide scale, which could undeniably help in advancing the knowledge of functional genomics, particularly in the construction of genetic interaction maps. In my opinion, the manuscript has met both the scientific and ethical standards for inclusion in the publication of scientific record. However, I have outlined few comments below that could help improve the quality of the manuscript. Comments: 1. P2/36. L75. The word “the” Consider removing one. 2. P6/36. L235. The word figure should be written with initial capital letter to make it consistent with the other usage in the manuscript. We thank the reviewer for their interest in our work and are grateful for their careful reading of our manuscript. We have removed the duplicate word “the” from L75 and replaced the word “figure” with “Fig” to adhere to the PLOS ONE guidelines for figures. Other changes to the manuscript:

1. Changed the in-text reference “Figure 6” to Fig 5 on L351.

2. Removed the subsection number that appeared before the subsection title on L360-L361.

3. Added “BUD4 from S288C” to the genotype of strain yWT01 in Table S1.

4.

Added the labels "a)" and "b)" to the panels of figure Fig7

---

## [Editor Report · Decision Letter 1]

8 Oct 2024

Saturated Transposon Analysis in Yeast as a One-step Method to Quantify the Fitness Effects of Gene Disruptions on a Genome-Wide Scale

PONE-D-24-01082R1

Dear Dr. LAAN,

We’re pleased to inform you that your manuscript has been judged scientifically suitable for publication and will be formally accepted for publication once it meets all outstanding technical requirements.

Kind regards,

Bashir Sajo Mienda, PhD

Academic Editor

PLOS ONE
---

## [Editor Report · Acceptance letter]

6 Nov 2024

PONE-D-24-01082R1 

PLOS ONE

Dear Dr. Laan, 

I'm pleased to inform you that your manuscript has been deemed suitable for publication in PLOS ONE. Congratulations! Your manuscript is now being handed over to our production team.

Kind regards, 

on behalf of

Dr. Bashir Sajo Mienda 

Academic Editor

PLOS ONE